# Low Rank Weight Bases for Visual Analogies

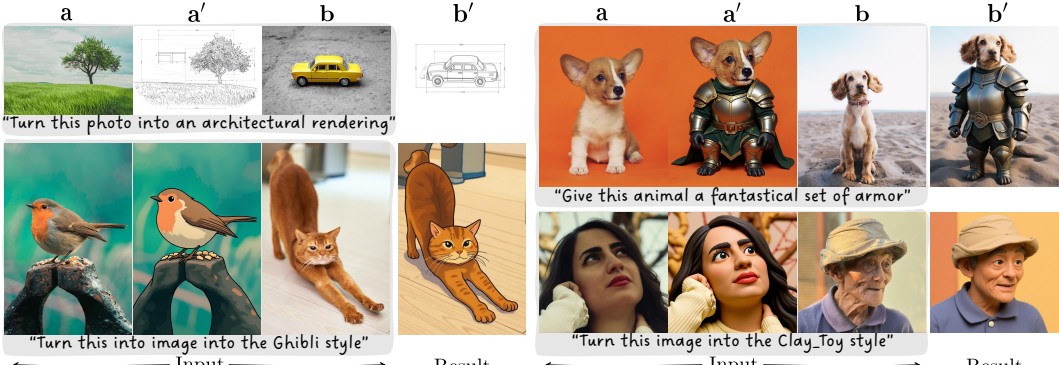

Figure 1: **LoRBA.** We present a novel method for analogy-based editing, based on learnable mixing of low-rank adapters. Given a prompt and an image triplet $\{\mathbf{a}, \mathbf{a}', \mathbf{b}\}$ that visually describe a desired transformation, LoRBA dynamically constructs a single LoRA from a learnable basis of LoRA modules, and produces an editing result $\mathbf{b}'$ that applies the same analogy for the new image.

## Abstract

Visual analogy learning enables image manipulation through demonstration rather than textual description, allowing users to specify complex transformations that are difficult to articulate in words. Given a triplet $\{\mathbf{a}, \mathbf{a}', \mathbf{b}\}$, the goal is to generate $\mathbf{b}'$ such that $\mathbf{a} : \mathbf{a}' :: \mathbf{b} : \mathbf{b}'$. Recent methods adapt text-to-image models to the analogy task using a single Low-Rank Adaptation (LoRA) module, but they face a fundamental limitation: attempting to capture the diverse space of visual transformations within a fixed adaptation module constrains generalization capabilities. Inspired by recent work showing that LoRAs in constrained domains span meaningful semantic spaces that can be interpolated, we propose LoRBA, a novel approach that specializes the model to each analogy task at inference time through dynamic composition of learned transformation primitives, informally, choosing a point in a "*space of LoRAs*". We introduce two key components: (1) a learnable basis of LoRA modules, to span the space of different types of visual transformations, and (2) a lightweight encoder that dynamically selects and weighs these basis LoRAs based on the specific analogy pair. Through comprehensive evaluations, we demonstrate that our approach achieves state-of-the-art performance and significantly improves generalization to unseen visual transformations. Our findings suggest that LoRA basis decompositions are a promising direction for flexible visual manipulation tasks.

## 1 Introduction

Text-based image editing models (Black Forest Labs et al., 2025; Brooks et al., 2023; Sheynin et al., 2024; Xiao et al., 2025; Zhang et al., 2025) have recently emerged as a powerful tool for controllable image generation and manipulation, enabling users to modify images through textual descriptions. However, many visual transformations are inherently difficult to articulate precisely through text alone. For example, consider describing the transformation that converts a photograph into the style of a specific painting, or conveying an exact target pose through text. Such inherent limitation motivates alternative paradigms that can capture and apply complex visual transformations.

Visual analogy learning (Hertzmann et al., 2001) offers a compelling solution to this challenge by enabling models to understand transformations through examples rather than explicit descriptions. In this paradigm, given a triplet of images $\{\mathbf{a}, \mathbf{a}', \mathbf{b}\}$, the goal is to generate an image $\mathbf{b}'$ such that the visual relationship $\mathbf{a} : \mathbf{a}' :: \mathbf{b} : \mathbf{b}'$ holds. That is, the transformation applied between $\mathbf{a}$ and $\mathbf{a}'$ should be analogously applied to $\mathbf{b}$ to produce $\mathbf{b}'$. This approach allows users to specify complex visual changes through demonstration, making it possible to capture nuanced transformations that would be difficult or impossible to describe textually.

Early learning-based approaches trained stand-alone analogy models directly from analogy data (Bar et al., 2022; Liu et al., 2024; Reed et al., 2015; Wang et al., 2023a;b; Yang et al., 2023), but this lead to limited task diversity and image quality, or required extensive compute. More recent work aims to leverage the rich prior of powerful text-to-image backbones by adapting them to the visual analogy task, using a single Low-Rank Adaptation (LoRA) module (Gong et al., 2025; Lu et al., 2025; Song et al., 2024). While effective, these methods face a fundamental limitation: they attempt to capture the diverse space of possible transformations within a single adaptation module. This constraint may limit the model's ability to generalize across the rich variety of relationships that exist in images.

We hypothesize that specializing the model to each specific analogy task at inference time may improve performance and generalization. While this objective could theoretically be achieved via hypernetworks that generate task-specific LoRAs (Song et al., 2024), these are notoriously difficult to train and often suffer from instability (Ortiz et al., 2024). Instead, we draw inspiration from recent work demonstrating that LoRAs from fine-tuned models (*e.g.*, for personalization tasks) tend to span a meaningful semantic basis, and that interpolating between these LoRAs can effectively cover new points in this semantic space (Dravid et al., 2024). Building on this insight, we explore a similar principle for visual analogy learning and propose LoRBA, a two-component system: (1) a learnable basis of LoRA modules and (2) a lightweight encoder that dynamically combines LoRAs from this basis at inference time based on the input analogy pair. These components are jointly trained, enabling the model to compose appropriate transformations for novel analogies unseen during training.

Existing methods typically encode analogy images using vision-language models such as CLIP (Radford et al., 2021) or SigLIP (Zhai et al., 2023) and provide these encodings as context to the generative model. This can provide the higher-level semantic understanding needed for understanding the analogy task. However, this might lead to loss of detail in fine-grained visual detail preservation. Recent advances have demonstrated that diffusion models can extract remarkably accurate visual details through extended attention mechanisms (Black Forest Labs et al., 2025; Cao et al., 2023). Thus, we leverage this capability by providing the full analogy triplet directly to the diffusion model through an extended-attention mechanism, while reserving CLIP-based encodings specifically for LoRA selection. This approach allows LoRBA to strike a balance between the consistency of fine-details and the higher-level semantics required to understand the analogy task.

We evaluate LoRBA against established baselines and demonstrate it achieves state-of-the-art results. Our contributions include: (1) a novel architecture that decomposes visual analogy learning into a basis of LoRAs with dynamic composition, and (2) a comprehensive evaluation showing improved generalization to unseen transformations compared to existing single-LoRA approaches.

## 2 RELATED WORK

**Visual analogies.** Visual analogies, also commonly referred to as "Image Analogies" (Hertzmann et al., 2001), "Visual Prompting" (Bar et al., 2022) or "Visual Relations" (Gong et al., 2025), is the task of learning a transformation from a pair of before-and-after exemplars and applying it analogously to new images. Early non-neural methods learned explicit per-pair filters for simpler tasks such as style transfer (Hertzmann et al., 2001). With the advent of network-based methods, initial works proposed models conditioned on an image embedding space where analogies can be presented through simple vector arithmetic (Reed et al., 2015). While these methods showed promise on datasets of simple, isolated objects, they struggled with the complexity of real-world images. Newer methods instead phrase the analogy task as one of in-context learning, where the model is directly conditioned on the exemplar pair and a reference image, and is trained to successfully synthesize the matching target (Bar et al., 2022; Wang et al., 2023a;b; Yang et al., 2023). More recently, some works build on pre-trained text-to-image foundation models and adapt them to the new task using a LoRA module (Chen et al., 2025; Gong et al., 2025; Hu et al., 2022). Although these methods show impressive results, they still struggle with generalization to unseen tasks. Our approach

aims to tackle this limitation by avoiding the bottleneck of a single LoRA, opting instead to train a basis of adapters which can be mixed in order to achieve greater flexibility and better generalization.

**Diffusion-based image editing.** The unprecedented semantic control offered by large scale text-to-image diffusion models (Black Forest Labs et al., 2025; Ramesh et al., 2022; Rombach et al., 2022) has inspired extensive work leveraging them as priors for image editing. Early works add noise to an image and remove it conditioned on a novel prompt (Meng et al., 2022), though such methods significantly change image structure. To better preserve the original content, more advanced approaches manipulate internal feature representations (Hertz et al., 2022; Parmar et al., 2023; Tumanyan et al., 2022) or consider the denoising trajectory of the model (Deutch et al., 2024; Hertz et al., 2023a; Huberman-Spiegelglas et al., 2023; Kulikov et al., 2024). Recent works go beyond text-only control and incorporate different control modalities for enhanced precision, such as ControlNet (Cao et al., 2023; Zhang et al., 2023), or attention-sharing (Alaluf et al., 2023; Gal et al., 2024; Hertz et al., 2023b; Tewel et al., 2024b). Additional work explores editing without text to enable modifications that cannot be textually described (Haas et al., 2024; Manor & Michaeli, 2024), though these methods only allow exploration without direct control of the results. With transformer-based diffusion models, attention-sharing approaches have gained popularity for maintaining subject consistency in personalization (Gal et al., 2023; Ruiz et al., 2022) and editing (Cai et al., 2025; Tan et al., 2025; Tewel et al., 2024a). Among these, Flux.1-Kontext (Black Forest Labs et al., 2025) was specifically trained for text-based image editing, incorporating input images through extended attention mechanisms. Our work extends this model's capabilities to visual analogies.

**LoRA and weight bases.** LoRA (Hu et al., 2022) is a parameter-efficient fine-tuning method that modifies a model through a series of low-rank matrices learned on top of the existing weights. Its success has lead to a range of downstream approaches trying to improve on the original formula. Of these, a line of work explores the combination of multiple LoRa modules, either to combine them post-tuning (Shah et al., 2024; Zhang & Xiong, 2025), or as a means of turning an existing model into a mixture of experts (Feng et al., 2024; Mao et al., 2025; Wu et al., 2024). In visual content generation, a recent work (Dravid et al., 2024) showed that independently trained LoRA weights can span a semantic basis, and interpolations between them can be meaningful. Similar observations were made in language processing, where LoRAs were combined for tasks like text simplification across different scientific domains (Cheng et al., 2025). We propose to further expand on this idea by learning a joint basis of LoRAs, along with the router to mix and match between them. Thus, we can learn a base that is more amenable to interpolations, and enable better downstream generalization.

## 3 METHOD

### 3.1 PRELIMINARIES

**Low-rank adaption.** LoRA (Hu et al., 2022) offers a parameter-efficient alternative to conventional fine-tuning of large models by learning low-rank matrices that adapt the pre-trained weights. Specifically, starting from a frozen pre-trained weight matrix $\boldsymbol{W}_0 \in \mathbb{R}^{m \times n}$, the update to the weights is represented as the product of two learned low-rank matrices $\Delta \boldsymbol{W} = \boldsymbol{BA}$, where $\boldsymbol{B} \in \mathbb{R}^{m \times r}$ and $\boldsymbol{A} \in \mathbb{R}^{r \times n}$, and the rank $r$ is typically $r \ll \min(m, n)$. This formulation drastically reduces the number of trainable parameters, while typically maintaining model performance. The final weights of the model are then updated to $\boldsymbol{W} = \boldsymbol{W}_0 + \frac{\alpha}{r} \boldsymbol{BA}$, where $\alpha$ is a scaling constant.

**Flow models.** Flow-based generative models (Albergo & Vanden-Eijnden, 2023; Lipman et al., 2023; Liu et al., 2023) learn a series of transformations to map samples from one probability distribution $\mathbf{x}_1 \sim p$, to samples from another $\mathbf{x}_0 \sim q$. In the generative context, $p$ is typically taken as the standard normal distribution, while $q$ is the data distribution in a latent space (Rombach et al., 2022). Then, these models learn a time-dependent velocity field $v_\theta(\mathbf{z}_t, t)$ that models the direction from a noisy sample towards the data manifold. The noisy sample $\mathbf{z}_t$ is a linearly interpolated latent between the two data distributions, $\mathbf{z}_t = (1 - t)\mathbf{x}_0 + t\mathbf{x}_1$. The rectified flow-matching training loss for a conditional models is then given as:

$$\mathcal{L} = \mathbb{E}_{t \sim p(t), \mathbf{x}_0, \mathbf{x}_1, \mathbf{y}, c} \left[ \|v_\theta(\mathbf{z}_t, t, \mathbf{y}, c) - (\mathbf{x}_1 - \mathbf{x}_0)\|_2^2 \right]. \tag{1}$$

Here, the velocity field is optionally conditioned on a context image $\mathbf{y}$, and a text-prompt $c$.

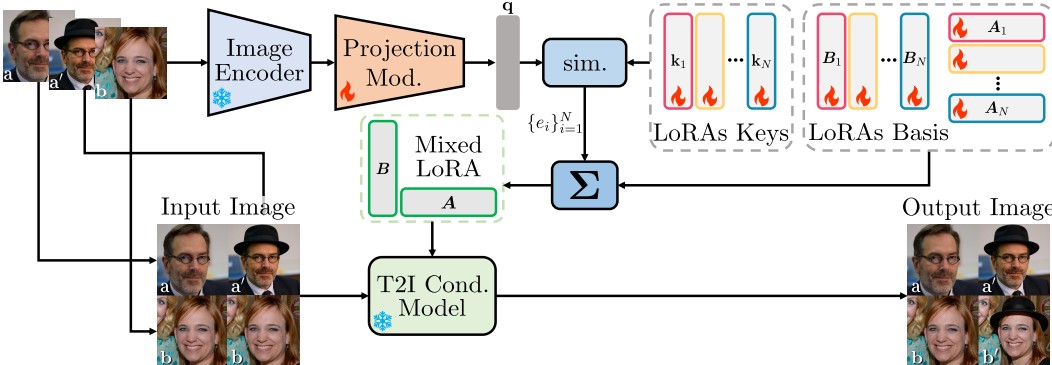

Figure 2: **LoRBA Overview.** We first encode $\mathbf{a}$ and $\mathbf{a}'$, that describe a visual transformation (*e.g.* adding a hat to the man), and $\mathbf{b}$, which should be edited analogously (*e.g.* adding a hat to the woman) with CLIP (Radford et al., 2021), and a small learned projection module. The similarity between the encoded vector and a set of learned keys determines the linear coefficients for combining the learned LoRAs into a single, mixed LoRA. This mixed LoRA is injected into a conditional flow model (*e.g.* Flux.1-Kontext (Black Forest Labs et al., 2025)). Next, we build a $2 \times 2$ composite image from $\{\mathbf{a}, \mathbf{a}', \mathbf{b}\}$. The conditional flow model gets this composite image as its input, along with a guiding edit prompt, and produces a composite image with the edited results $\mathbf{b}'$ in the bottom-right quadrant.

## 3.2 LoRBA

Our objective is to perform visual analogy completion (Hertzmann et al., 2001), where the model infers a proposed edit from a given pair of images and applies it to a new image. Formally, two reference images, $\mathbf{a}, \mathbf{a}' \in \mathbb{R}^D$, are related by some unknown transformation $\mathcal{T} : \mathbb{R}^D \to \mathbb{R}^D$ such that $\mathbf{a}' = \mathcal{T}(\mathbf{a})$. Given a new image $\mathbf{b} \in \mathbb{R}^D$, the goal is to generate $\mathbf{b}' \in \mathbb{R}^D$ such that $\mathbf{b}' \approx \mathcal{T}(\mathbf{b})$.

**Naive solutions and limitations.** Using a pre-trained conditional generative model, such as FLUX.1-Kontext (Black Forest Labs et al., 2025), existing solutions for this task fine-tune the model using a single LoRA (Ryu, 2023). For example, given $\{\mathbf{a}, \mathbf{a}', \mathbf{b}\}$, one can construct a composite $2\times 2$ image $\mathbf{y} = [\mathbf{a}, \mathbf{a}'; \mathbf{b}, \mathbf{b}]$, as shown in the bottom-left part of Fig. 2, which serves as the conditioning input. The goal of the model is to output $\mathbf{x}_0 = [\mathbf{a}, \mathbf{a}'; \mathbf{b}, \mathbf{b}']$, such that the bottom-right quadrant was transformed from $\mathbf{b}$ to $\mathbf{b}'$, by training over Eq. (1). While these approaches perform well when the transformation $\mathcal{T}$ is constrained to the training set's analogy types, they struggle to generalize to new, diverse transformations. We propose that this arises in part because the single adapter struggles to capture the wide range of analogical relationships, from different style transfers to objects insertion or layout modifications.

A more advanced solution could then be to span the diverse set of possible analogies using multiple adapters. In a recent work, Dravid et al. (2024) demonstrated that LoRAs trained for model personalization can span a semantic basis. Inspired by their work, we propose to learn such a basis for *task LoRAs*. A naïve adaptation of Dravid et al. (2024) to the analogy tasks would require us to first optimize a single adapter for each of $N$ analogy types seen during training, such that each LoRA module $i$ excels at a different subset of visual edits. Once the specialized adapters are trained, they can be linearly combined to obtain an equivalent single "novel" adapter

$$\mathbf{A} = \sum e_i \mathbf{A}_i, \quad \mathbf{B} = \sum e_i \mathbf{B}_i, \tag{2}$$

where the coefficients $e_i$ are optimized for each analogy task through the use of Eq. (1) and $\{\mathbf{a}, \mathbf{a}'\}$. The model using the combined LoRA is then used to transform $\mathbf{b}$ to $\mathbf{b}'$.

However, this approach requires training a large number of models, and a test-time tuning phase for every new analogy. Indeed, Dravid et al. (2024) required $65,000$ LoRAs to capture the constrained space of faces, and collecting a significant number of different analogy pairs is more difficult.

**Our appraoch.** Instead, we propose LoRBA. Rather than training individual LoRAs and combining them only at inference time, we propose to simultaneously train a basis of LoRA adapters, jointly

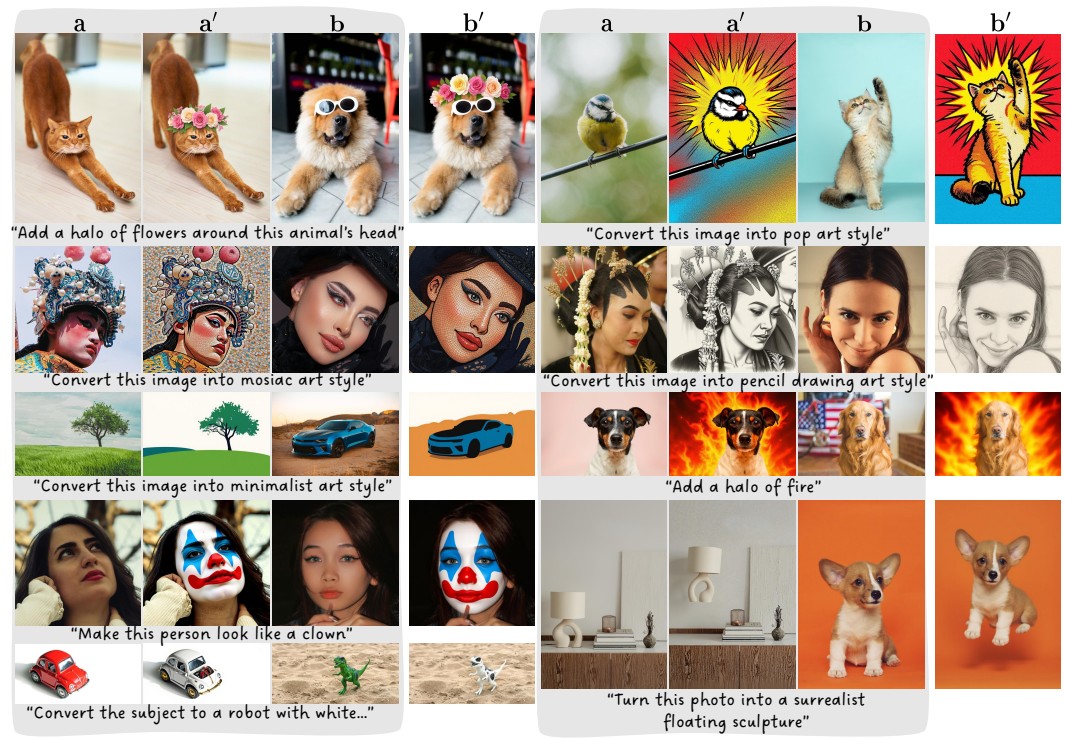

Figure 3: **LoRBA visual analogy results.** The use of a LoRA Basis allows LoRBA to generalize to a wide varity of new analogy tasks, from adding objects such as a crown of flowers, transfering specific styles or makeup, and copying pose changes. Please zoom in for more details.

with an encoder that predicts linear-combination coefficients for each input analogy pair. Specifically, we maintain a set of $N$ rank-$r$ LoRAs, and associate each $\boldsymbol{A}_i, \boldsymbol{B}_i$ pair where $i \in \{1, \ldots, N\}$ with a learnable key vector $\mathbf{k}_i \in \mathbb{R}^d$, as depicted in the right part of Fig. 2. We additionally define an encoder network based on a frozen, pre-trained ViT (Zhai et al., 2022), $\mathcal{E}$, such as CLIP (Radford et al., 2021). The encoder network takes as input the conditioning image triplet, $\{\mathbf{a}, \mathbf{a}', \mathbf{b}\}$, passes them through the pre-trained ViT, concatenates the results and project them through a small learnable projection module $\mathcal{P}$ that outputs the results as a query vector $\mathbf{q} \in \mathbb{R}^d$:

$$\mathbf{q}(\mathbf{a}, \mathbf{a}', \mathbf{b}) = \mathcal{P}\Big(\big[\mathcal{E}(\mathbf{a}), \mathcal{E}(\mathbf{a}'), \mathcal{E}(\mathbf{b})\big]\Big). \tag{3}$$

Then, based on the conditioning query, we compute $N$ coefficients with

$$e_i(\mathbf{a}, \mathbf{a}', \mathbf{b}) = \left[\text{softmax}\left(\frac{\mathbf{q}(\mathbf{a}, \mathbf{a}', \mathbf{b})\boldsymbol{K}^T}{\sqrt{d}}\right)\right]_i, \tag{4}$$

where $K \in \mathbb{R}^{d \times N}$ contains the key vectors $\{\mathbf{k}_i\}_{i=1}^N$ in its columns. The final LoRA combination follows Eq. (2). and is marked as "Mixed LoRA" in Fig. 2.

We use the same pre-trained encoder across different network layers, but train individual LoRBA modules, including LoRAs, keys and projection modules for each targeted weight matrix $\boldsymbol{W}_0$ in the network. This enables capturing different semantic elements for each weight and layer in the model.

## 4 EXPERIMENTS

**Settings.** We evaluate our approach using Flux.1-Kontext (Black Forest Labs et al., 2025) as the pre-trained conditional flow model and CLIP (Radford et al., 2021) as the image encoder backbone. For our LoRAs Basis, we match the capacity of prior work (Gong et al., 2025), using $N = 32$ adapters, each of rank $r = 4$, with $d = 128$ as the learned key dimension. We project the CLIP-encoder's output to $\mathbb{R}^d$ using a single fully-connected layer. To save on compute, we set the training

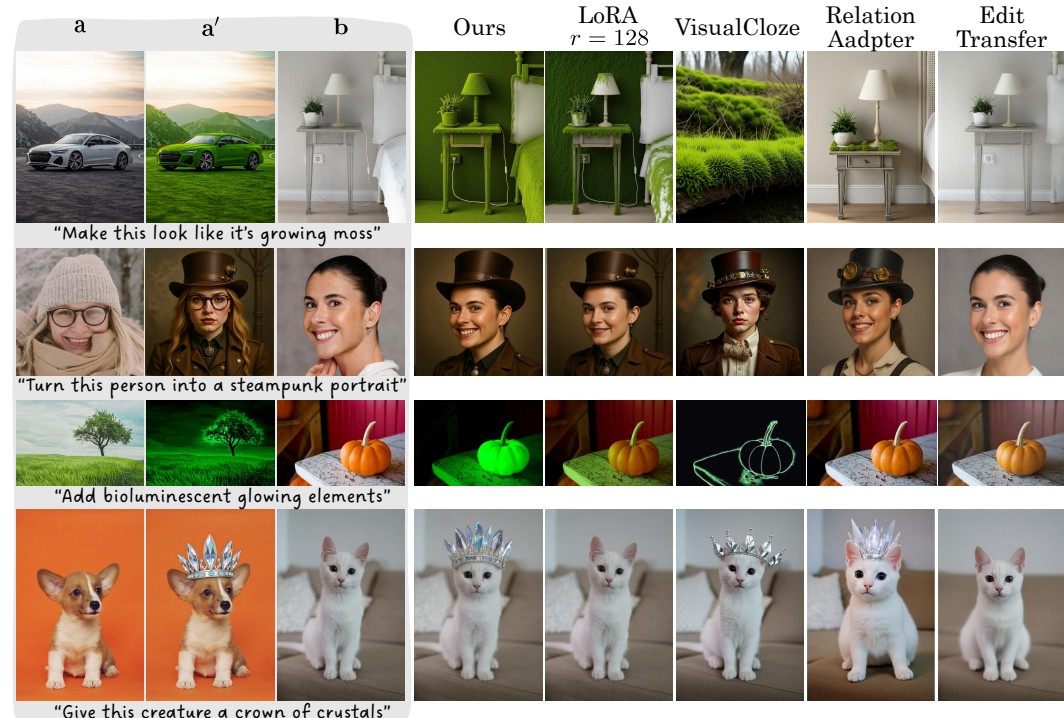

Figure 4: **Comparisons with baseline methods on unseen tasks.** Our approach generalizes across more diverse tasks, and better maintains the visual details of both the subject and the analogy.

resolution to a maximum of $512 \times 512$ images, resizing on the long-edge of images if necessary. Additional implementation details are in App. A. We compare LoRBA to four baselines: A standard LoRA of similar total parameter capacity (equivalent to LoRBA with $N = 1, r = 128$), trained on top of Flux.1-Kontext, as well as three prior visual analogy methods based on Flux.1-Dev (Relation-Adapter (Gong et al., 2025), VisualCloze (Li et al., 2025) and EditTransfer (Chen et al., 2025)).

**Dataset.** We train our model using the public Relation252k (Gong et al., 2025) set, which contains 16K analogy image pairs across 208 tasks. Since the test set of Relation252k is not publicly available, we create a custom validation set to evaluate visual analogies. Specifically, we focus on analogies which were not found in the training set, which we create in the following manner: First, we collect over 100 photos from Unsplash[1] covering diverse concepts from three categories: animals, persons, and general objects. Next, we create analogy pairs with a focus on two categories: transformations which are in-domain for the base text-to-image model, and transformations that are not. For in-domain transformations, we first use an LLM to summarize the training prompts for each task in the training-set of Relation252k, yielding 208 representative prompts. Next, we ask the LLM to generate novel prompts that differ from the training set's prompts and manually verify that they match the given concept categories. We filter prompts on which Flux.1-Kontext fails to produce a meaningful edit, and further randomly select 15 prompts per concept category from the remainder. We generate three random images per prompt, obtaining a total of 135 analogy pairs. For out-of-domain analogies, we collect 18 community LoRAs for Flux.1-Kontext from HuggingFace, which were trained to enable edits that the base model failed with. We use these pre-trained LoRAs, and repeat the previous random sampling strategy to get 135 analogy pairs. Finally, we randomly select for the input images **b** two images from the matching concept category with a similar aspect ratio, cropping them to the exact size of **a** and **a′**. In total, our set contains 540 analogy triplets across 90 tasks and 3 concept categories. Additional details on our validation set can be found in App. A.

---

[1] https://unsplash.com/

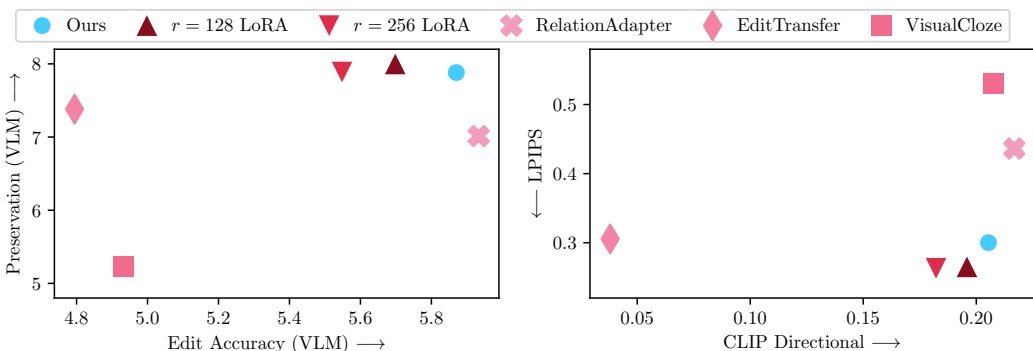

Figure 5: **Quantitative comparisons.** (left) Accuracy of the applied edit and preservation of **b** in **b** using Gemma-3 (Team et al., 2025). Top right is better. (right) CLIP directional similarity and LPIPS between **b**′ and **b**. Bottom-right is better. Our method pushes the Pareto front of edit accuracy-preservation, achieving higher edit accuracy while strongly preserving the input image.

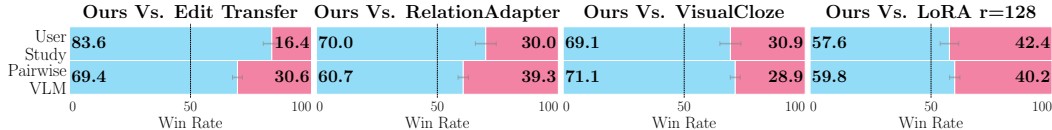

Figure 6: **Pairwise image comparisons.** We compare LoRBA to 4 baselines on overall edit quality preference via both a user study and using a VLM. LoRBA produces edits that are favored by both. Error bars are the 68% Wilson score interval.

### 4.1 QUALITATIVE EVALUATIONS

Figures 1 and 3 include results of analogy-based editing using LoRBA. Notably, the model generalizes to new tasks covering style transfer, background replacements, object insertion, object displacement and more. In Fig. 4 we show qualitative comparisons of our method against the baselines. Notably, existing approaches either struggle with maintaining the content of the original image, or fail on some of the tasks. Our method meanwhile shows greater adaptability and succeeds in a wider range of unseen tasks. Additional results appear in App. B.

### 4.2 QUANTITATIVE EVALUATIONS

**Automated evaluation metrics.** For quantitative evaluations, we follow prior work (Chen et al., 2025; Gu et al., 2024; Song et al., 2024) and evaluate performance across standard metrics such as LPIPS (Zhang et al., 2018) between the source and generated image, and CLIP directional similarity between both analogy pairs. In addition, we build on recent image editing work (Huang et al., 2025), which demonstrates that VLMs often better correlate with human preference than CLIP-based methods, and implement a VLM-based assessment protocol. Specifically, we conduct two VLM-based experiments: In the first, we provide Gemma-3 (Team et al., 2025) with $\{\mathbf{a}, \mathbf{a}', \mathbf{b}, \mathbf{b}'\}$, and ask the VLM to evaluate the quality of results on two criteria: consistency with the source image, and accuracy of the applied transformation relative to the reference transformation. We name these metrics as *Preservation (VLM)* and *Edit Accuracy (VLM)*, respectively. As a second quality metric, we take a 2-alternative-forced-choice design (2AFC). We show Gemma-3 $\{\mathbf{a}, \mathbf{a}', \mathbf{b}\}$, the $\mathbf{b}'$ result of our model, and the $\mathbf{b}'$ result generated by one baseline, and ask it to select the image that better applies the analogy. We report this metrics as *Pairwise VLM*. The prompts given to the VLM and further details appear in App. A. The results are shown in Fig. 5 and Fig. 6. When considering preservation and editing accuracy tradeoffs (Fig. 5), our model pushes the Pareto front, achieving high edit accuracy while better maintaining the input's structure and appearance.

Table 1: Ablation study of LoRBA for different hyperparameter and architecture choices

| Model | Pres. ↑ (VLM) | Acc. ↑ (VLM) | LPIPS ↓ | CLIP Dir. ↑ | Pairwise VLM (%) ↑ | | | |
|---|---|---|---|---|---|---|---|---|
| | | | | | LoRA $r = 128$ | ET | VC | RA |
| LoRBA (full) | 7.88 | 5.87 | 0.30 | 0.21 | 59.8 | 69.4 | 71.1 | 60.7 |
| $+ \, r = 16$ | 8.17 | 4.78 | 0.17 | 0.10 | 51.9 | 60.9 | 64.4 | 50.2 |
| $+ \, r = 16, N = 8$ | 7.85 | 5.36 | 0.27 | 0.181 | 59.8 | 70.7 | 69.3 | 59.3 |
| $+$ Tanh | 7.97 | 4.41 | 0.16 | 0.07 | 51.9 | 55.4 | 55.2 | 46.7 |
| $+ \, 2 \times 2$ Inp. | 7.94 | 5.65 | 0.27 | 0.19 | 62.6 | 73.0 | 71.1 | 55.6 |
| $+$ SL2 | 7.85 | 5.73 | 0.30 | 0.20 | 58.9 | 68.9 | 75.9 | 56.3 |
| $+$ SL2, $2 \times 2$ Inp. | 7.9 | 5.63 | 0.28 | 0.19 | 59.8 | 74.3 | 72.4 | 61.1 |

**User study.** Beyond automated metrics, we also conduct a two-alternative forced choice user study. We show each user a reference pair $(\mathbf{a}, \mathbf{a}')$, an input image $\mathbf{b}$, and two results (one from our model and one of a random baseline). Users are asked to select their preferred editing result. In total, we collected responses from 33 users covering 45 image pairs. The results (Fig. 6) align with the automated metrics, showing that users favor our approach over all baselines.

All in all, our experiments demonstrate that our approach can meaningfully improve on the existing state of the art, and better generalize to unseen tasks.

### 4.3 ABLATIONS

We next study the importance of different components of LoRBA.

**Basis of LoRAs advantages.** While our approach showed improved performance compared to a standard LoRA, it introduces additional parameters via projection modules and learned keys. Hence, we first validate LoRBA's performance against a LoRA with higher capacity, of $r = 256$. As shown in Fig. 5, naïve parameter addition does not strictly correlate with better performance.

**Capacity effect.** Similarly, we compare LoRBA across modified capacities in both basis sizes $N$ and ranks $r$. Specifically, we compare our original variation ($\{N = 32, r = 4\}$), with $\{N = 16, r = 16\}$ and $\{N = 32, r = 16\}$. We use the same evaluation setup as the quantitative comparisons. Results are reported in Tab. 1. Reducing the basis size while maintaining the capacity ($r = 16, N = 8$) leads to a slight drop in performance. This highlights the importance of a large basis for generalization. Similarly, a naïve increase in rank can hamper editability, which we hypothesize to be a consequence of the data, leading to increased overfitting.

**Similarity normalizing function.** The normalization function choice in Eq. (4) can also affect the learned basis. For example, the used softmax is bound to $[0, 1]$, hence it cannot result in negative coefficients for any LoRA. An alternative approach is to use Tanh, which is instead bound to $[-1, 1]$. In practice, we find it to drastically underperform. We propose that this may be due to Tanh allowing the model to compose mixed LoRAs with much greater norms, possibly taking the model too far out of domain. However, we leave further investigation of activations to future work.

**Layout of encoder input.** In our approach, we elected to separately encode each of the conditioning analogy images using CLIP, and concatenate their representations. Our intuition is that CLIP requires resizing the image to $224 \times 224$, which can severely constrain the level of detail in each quadrant of the $2x2$ grid that we provide Flux as a context. Moreover, concatenated features could allow the model to better understand which encoding represents each conditioning image (*i.e.* $\mathbf{a}$, $\mathbf{a}'$ and $\mathbf{b}$), allowing it to better reason over the analogy. We verify this experimentally by comparing to a version that provides CLIP with just the context image (the $2 \times 2$ grid). As seen in Tab. 1, this diminishes results, mainly decreasing the editing-accuracy metrics.

**Alternative image encoders.** Although our approach uses CLIP (Radford et al., 2021) as an encoder backbone, we validate our robustness to alternative, common choices, and specifically SigLIP2 (Tschannen et al., 2025). The results in Tab. 1 indicate that changing the encoder does not significantly alter our performances. We leave further tuning of encoders to future work.

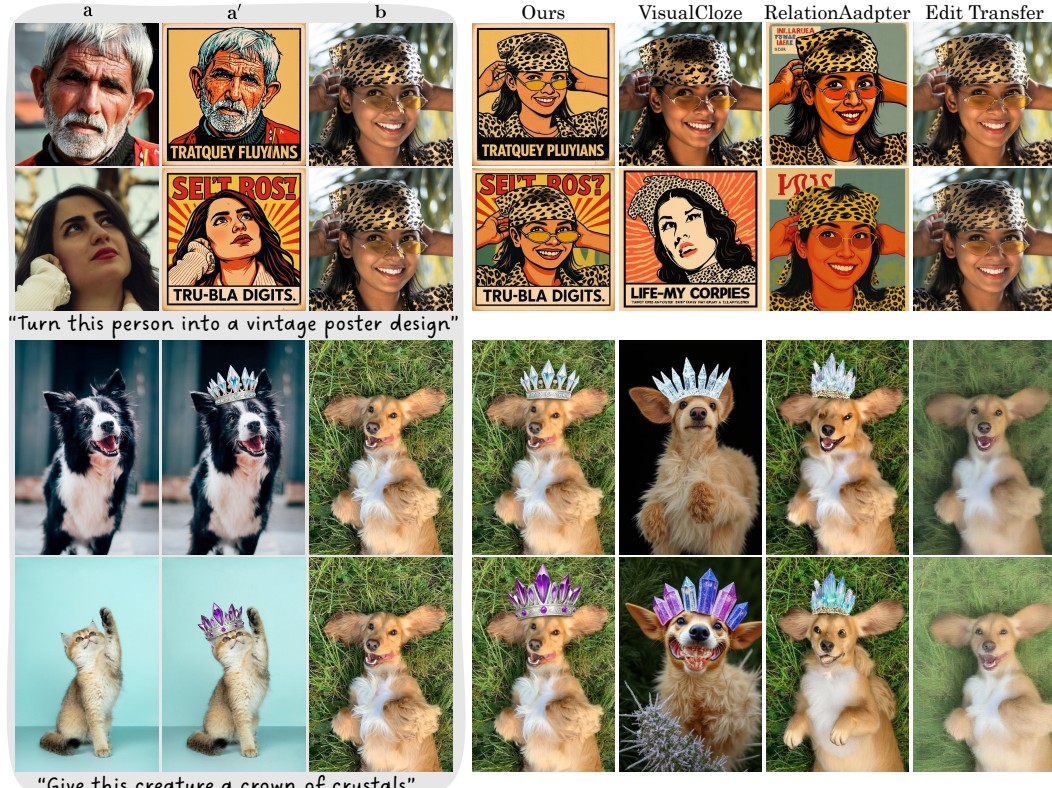

Figure 7: **Effect of different reference analogy pairs.** LoRBA directly leverages the analogy pair to understand the details of the proposed task, applying an edit that is beyond just text-based editing based on the given prompt. For example, when the prompt is "Give this creature a crown of crystals", the analogy context passes information on the amount and color of the crystals.

**Importance of prompts and reference images.** We follow existing baselines and use prompts to augment the model's understanding. Since our goal is analogy based editing, and not simply text-based modification, we verify that our model is indeed affected by the choice of analogy pair. Specifically, we examine how the same input image, $\mathbf{b}$, reacts to different reference pairs $\{\mathbf{a}, \mathbf{a}'\}$ under the same editing prompt. As can be seen in Fig. 7, the reference pair dictates the details of the analogy task, and particularly the visual details that are not captured by the prompts. For example, it can copy the text of the analogy image, adapt its specific style, or match the design and colors of the given crown. In comparison, we observe that some of the baselines are insensitive to the analogy pair, instead relying almost entirely on the prompt. As this experiment demonstrates, our approach has learned to perform analogy-based editing, and to a greater degree than the existing baselines.

## 5 DISCUSSION

We introduced LoRBA, a modular framework for visual analogy completion that learns a basis of LoRA adapters and dynamically composes them using a shared encoder conditioned on the input analogy. Our approach addresses the limitations of single-adapter fine-tuning or multi-adapters optimization at inference time by enabling flexible, layer-specific adaptations to diverse and unseen transformations. Through structured composition, we showed how LoRBA outperforms and generalizes better than competing naive LoRA-based methods across various visual analogy tasks. However, this generalization is not without limits. For example, LoRBA may still struggle with tasks that are significantly different from the training corpus. While our focus in this work is on analogy completion, the LoRA-basis approach could be broadly applicable, possibly replacing LoRAs in other tasks where generalization is needed. We hope to explore this direction in future work.

## REPRODUCIBILITY STATEMENT

Our code will be published upon acceptance, including all training and evaluation scripts. We detail all needed implementation details for reproducing the results and evaluations in App. A.

## ETHICS STATEMENT

This work aims to advance the field of image editing and in particular image editing via analogies. While our method enables flexible visual transformation modeling, we acknowledge the potential for misuse in generating misleading or deceptive content. We use controlled analogy tasks and do not deploy our method in an open-ended generative settings and unsafe transformations. Nevertheless, users might use our methods to edit images of others into misleading context without permission, which is a common problem with regard to the entire image editing field. We therefore believe a crucial step in the field is developing reliable methods for automatic detection of AI-generated content, and specifically wether AI-methods were used in editing of images.

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

## A EXPERIMENTAL DETAILS

### A.1 IMPLEMENTATION DETAILS

In all our experiments, we train for 10K steps on 1 H100 GPU, setting 8-bit AdamW (Loshchilov & Hutter, 2019) as the optimizer with a learning rate of $10^{-3}$, $\beta_1 = 0.9$, $\beta_2 = 0.99$, a weight decay value of 0.05, and bfloat16 mixed-precision training. We enable gradient checkpointing, and use a batch size of 6 for all experiments, except for when $r = 16, N = 32$ where the batch size is set to 4. As for the encoders, the CLIP checkpoint we use is `openai/clip-vit-large-patch14`. For the SigLIP2 version in the ablations, we test `google/siglip2-base-patch16-224`. Both output a vector in $\mathbb{R}^{768}$.

### A.2 INFERENCE DATASET

All images from Unsplash are free to use under the Unsplash license[2]. To simulate in-domain prompts, we use GPT-4o (OpenAI, 2024) and Claude Sonnet 4 (Anthropic, 2025) to summarize the training prompts of Relation252k (Gong et al., 2025) as described in Sec. 4, and generate novel prompts. The 15 randomly selected prompts per concept category (animals, objects, and persons) appear in Tab. 2. The 18 pre-trained LoRA adapters are sourced from HuggingFace[3], and cover a range of transformation types such as style transfer, object modification, and artistic reinterpretation. Specifically, we use the following community LoRAs, with their provided trigger prompt:

- `day-dream/MechAnything-Kontext-Dev-Lora`
- `drbaph/Fluffy-kontext-LoRA`
- `fal/3D-Game-Assets-Kontext-Dev-LoRA`
- `fal/Cubist-Art-Kontext-Dev-LoRA`
- `fal/Gouache-Art-Kontext-Dev-LoRA`
- `fal/Minimalist-Art-Kontext-Dev-LoRA`
- `fal/Mosaic-Art-Kontext-Dev-LoRA`
- `fal/Pencil-Drawing-Kontext-Dev-LoRA`
- `fal/Plushie-Kontext-Dev-LoRA`
- `fal/Pop-Art-Kontext-Dev-LoRA`
- `fal/Watercolor-Art-Kontext-Dev-LoRA`
- `gokaygokay/Bronze-Sculpture-Kontext-Dev-LoRA`
- `gokaygokay/Low-Poly-Kontext-Dev-LoRA`
- `gokaygokay/Marble-Sculpture-Kontext-Dev-LoRA`
- `gokaygokay/Oil-Paint-Kontext-Dev-LoRA`
- `Kontext-Style/Clay_Toy_lora`
- `Kontext-Style/Ghibli_lora`
- `Kontext-Style/Paper_Cutting_lora` .

To match between $\mathbf{a}, \mathbf{a}'$ and $\mathbf{b}$ images of different sizes, we only choose $\mathbf{b}$ images with an original aspect ratio distanced 0.15 from the aspect ratio of $\mathbf{a}$ and $\mathbf{a}'$, and crop $\mathbf{b}$ to $\mathbf{a}$'s aspect ratio. The images are resized to the same size with a maximum long edge of 512 before entering Flux.1-Kontext.

---

[2]https://unsplash.com/license
[3]https://https://huggingface.co/

Table 2: List of prompts generated for the inference sets

| Category | Prompt |
|---|---|
| Animals | Add a collar with a bell |
| Animals | Add a mountainous background |
| Animals | Give this animal clockwork mechanical parts |
| Animals | Add a flowing mane |
| Animals | Add camouflage patterns |
| Animals | Give this animal ethereal ghost-like transparency |
| Animals | Add a flowing river background |
| Animals | Add metallic golden fur highlights |
| Animals | Give this animal translucent fairy wings |
| Animals | Add a halo of fire |
| Animals | Give this animal a fantastical set of armor |
| Animals | Give this creature a crown of crystals |
| Animals | Add a halo of flowers around this animal's head |
| Animals | Give this animal bioluminescent markings |
| Animals | Make this creature look sleepy |
| Objects | Add a swirling galaxy background |
| Objects | Render the object entirely as if it's made from hand-knitted or hand-crocheted yarn |
| Objects | Add bioluminescent glowing elements |
| Objects | Turn this into a candy or confectionery version |
| Objects | Add flowing fabric or silk textures |
| Objects | Turn this into a steampunk mechanical design |
| Objects | Add intricate filigree patterns |
| Objects | Turn this into a vintage advertisement poster |
| Objects | Give this object a coat of rust |
| Objects | Turn this photo into a cross-section diagram |
| Objects | Make this look ancient and archaeological |
| Objects | Turn this photo into a surrealist floating sculpture |
| Objects | Make this look like it's growing moss |
| Objects | Turn this photo into an architectural rendering |
| Objects | Make this look like it's made of clouds |
| Persons | Add a cape or cloak |
| Persons | Add elaborate hairstyling with ornaments |
| Persons | Make this person look heroic |
| Persons | Add a serene, forested background |
| Persons | Add golden hour lighting to this portrait |
| Persons | Make this person look like a clown |
| Persons | Add a swirling vortex background |
| Persons | Add natural outdoor lighting to this portrait |
| Persons | Make this person look like royalty |
| Persons | Add body paint or decorative patterns |
| Persons | Add temporary tattoos |
| Persons | Turn this person into a holographic projection |
| Persons | Add elaborate eye makeup |
| Persons | Make this person look ethereal |
| Persons | Turn this person into a steampunk portrait |

## A.3 VLM BASED EVALUATION

Part of our automated evaluation metrics include the use of Gemma-3 ([Team et al., 2025](#)) as a VLM to evaluate our results. We use two VLM-based experiments. In the first, we ask the VLM to evaluate our results on two criteria: consistency with the source image $\mathbf{b}$ and accuracy of the applied transformation relative to the reference transformation described by $\{\mathbf{a}, \mathbf{a}'\}$. For this, we provide Gemma-3 with $\{\mathbf{a}, \mathbf{a}', \mathbf{b}, \mathbf{b}'\}$, and the following prompt:

```
You are given 4 images:  A (original image), A' (edited version
of A), B (another original image), and B' (an output of an editing
method).  A, A' and B are reference images that are given to some
editing method in order to generate B'. The method tries to infer
the transformation that A underwent to produce A', and then tries
(maybe unsuccessfully) to apply the exact same transformation to B
- in order to generate B'. Your task is to evaluate the resulting
B': Was the same transformation applied well?
Specifically, assess B' under two metrics, editing accuracy, and
consistency with the original image B, 1-10 integers only:
1) editing accuracy:  Evaluate how closely B' applies the
transformation seen from A to A'. Are there missing elements, are
there redundant elements?  Quantify the precision of the editing.
2) consistency:  Asses how well the edited image B' maintains the
context of the original image B. Does it preserve the identity,
objects, and layout in B that did not require a change, based on
the infered transformation from A to A'?
Consider in your evaluations other visual factors such as the
localization of the edits, existence of redundant elements,
style/strength/magnitude/colors of changes.  First, describe in
detail what the transformation from A to A'. Then describe what
elements of it are present or missing in B', detailing precisely
what's wrong regarding each metric.
Then, return a strict JSON with this scheme:
{"metrics":{"accuracy":<1-10>,"consistency":<1-10>},
"explanation":"the reasoning you described above"}.
```

The JSON is parsed automatically, and we report the numeric values as *Preservation (VLM)* and *Edit Accuracy (VLM)*.

In the second quality metric, we take a 2-alternative-forced-choice design (2AFC). We show Gemma-3 five images: $\{a, a', b\}$, the $b'$ result of our model, and the $b'$ result generated by one baseline, and ask it to select the image that better applies the analogy via the following prompt:

```
 You are given 5 images:  A (original image), A' (edited version
of A), B (another original image), and 2 B' images (outputs of
2 editing methods).  A, A' and B are reference images that are
given to some editing method in order to generate B'. The methods
try to infer the transformation that A underwent to produce A',
and then tries (maybe unsuccessfully) to apply the exact same
transformation to B - in order to generate B'.
Your task is to evaluate the resulting B's:  In which of the two
methods was the same transformation applied well?
Specifically, assess B' under two metrics, editing accuracy, and
consistency with the original image B, 1-10 integers only:
1) editing accuracy:  Evaluate how closely B' applies the
transformation seen from A to A'. Are there missing elements, are
there redundant elements?  Quantify the precision of the editing.
2) consistency:  Asses how well the edited image B' maintains the
context of the original image B. Does it preserve the identity,
objects, and layout in B that did not require a change, based on
the infered transformation from A to A'?
Consider in your evaluations other visual factors such as he
localization of the edits, existence of redundant elements,
style/strength/magnitude/colors of changes.
First, describe in detail what the transformation from A to A'.
Then describe what elements of it are present or missing in B'1
and B'2, detailing precisely what's wrong regarding each metric.
Then, return a strict JSON with this scheme:  {"better":<1 or
2>,"explanation":"the reasoning you described above"}
```

The JSON is parsed and we report the resulting winrates as *pairwise VLM*.

## B ADDITIONAL RESULTS

We provide additional qualitative results of our method in Fig. 8, as well as more comparisons of our method to the 4 baselines from Sec. 4 in Fig. 9.

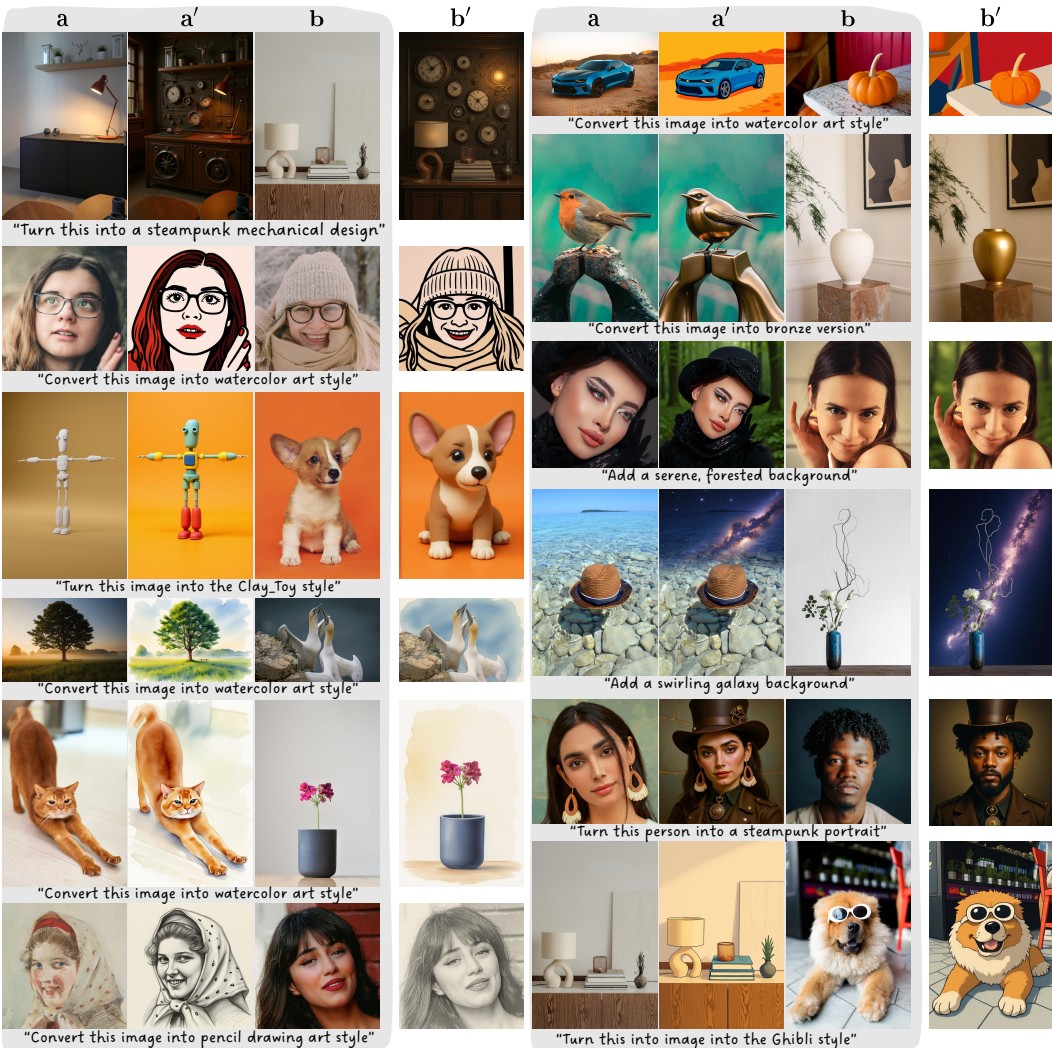

Figure 8: **LoRBA visual analoy results.** The use of a LoRA Basis allows LoRBA to generalize to a wide varity of new analogy tasks, from changing given images to certain styles such as clay toys or bronze sculptures, changing the backgrounds, or changing the cloths of the person. Please zoom in for more details.

## C LLM USAGE STATEMENT

LLMs such as GPT-4o (OpenAI, 2024), Claude (Anthropic, 2025) and CoPilot (Microsoft, 2025) assisted during the writing of this paper to refine the clarity and fluency of the text. In addition, as described in App. A, GPT-4o and Claude assisted in summarizing the prompts of Relation252K, as well as for generating novel prompts for evaluation. The Gemma-3 (Team et al., 2025) VLM was also used in our work, for evaluating the results, as described in Sec. 4 and App. A.

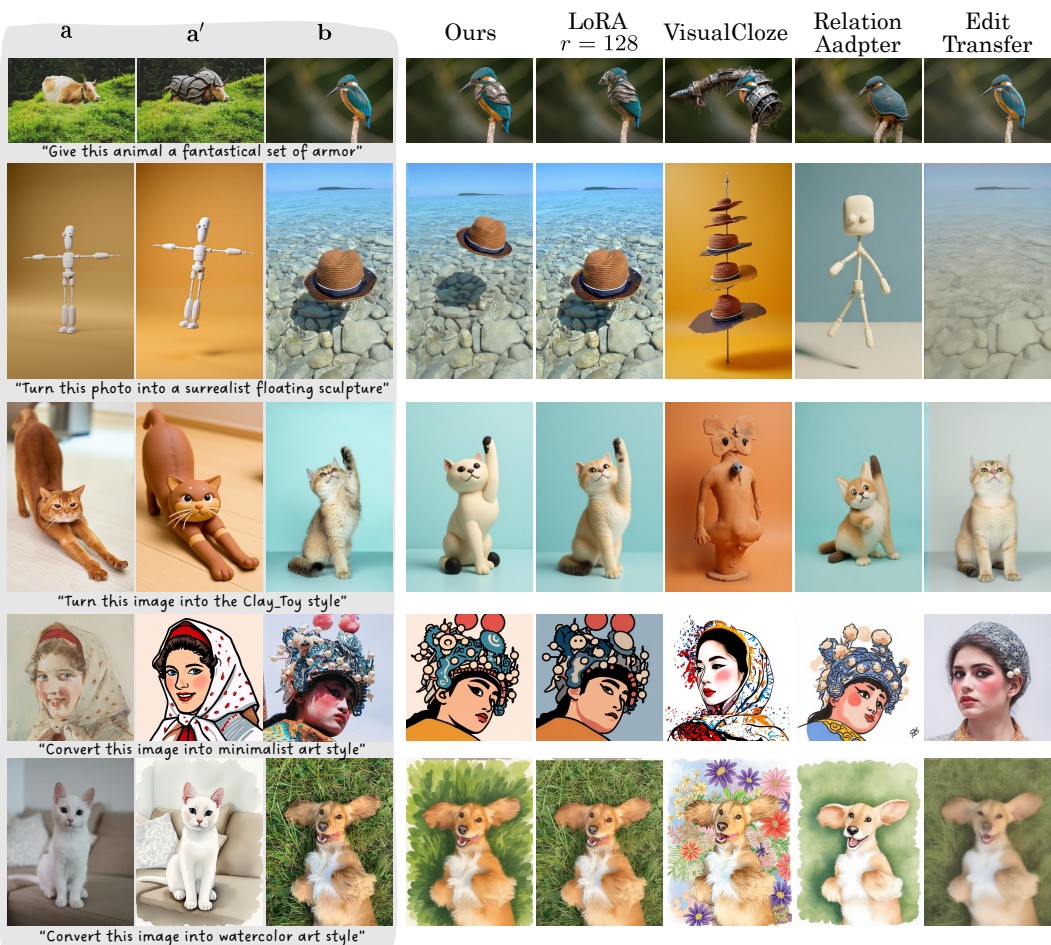

Figure 9: **Comparisons with baseline methods on unseen tasks**. Our approach generalizes more across diverse tasks, and better maintains the visual details of both the subject and the analogy.

