# OpenReview forum: "Low Rank Weight Bases for Visual Analogies"
_ICLR.cc/2026/Conference — ICLR 2026 Conference Withdrawn Submission_

### Official Review · Reviewer_9xKs · 2025-10-15

**Soundness:** 1
**Presentation:** 2
**Contribution:** 2
**Rating:** 2
**Confidence:** 2

**Summary:**

The authors propose a new approach, called LoRBA, which specializes in the model for image transformation tasks by leveraging the learned low-rank adapter parameters. The method follows a visual analogy framework, where the model is required to generate a new image by applying the same visual changes observed in a reference image pair to a target image. It contributes to improving the model’s generalization across diverse unseen image transformation tasks through a new architecture and provides detailed component-based analyses.

**Strengths:**

•	The paper provides a detailed ablation analysis that affects performance.

•	A variety of methods have been utilized to evaluate the results.

**Weaknesses:**

•	The paper employs the term “visual analogy” primarily in the context of style transfer, object addition, and related image transformation tasks. However, visual analogy refers to a broader process of relational knowledge transfer, extending beyond image manipulation.

•	The visual analogies were referred to as Visual Prompting and Visual Relations in Section 2. While these are related concepts, visual analogy generally encompasses a broader conceptual scope and is not strictly identical to the other two. It may be helpful for the authors to distinguish these terms more clearly.

•	The paper proposes to decompose visual analogy learning through the LoRBA architecture and includes edit prompts in experiments that describe the intended transformation. In visual analogy, however, the transformation is inferred from the relation between the reference pair (A → A′). When the text prompt explicitly defines this relation, the reasoning aspect of analogy is effectively replaced by instruction following. In that case, the task aligns more closely with prompt-conditioned image transformation that follows the analogy structure (A→A′ :: B→B′), but not the reasoning process itself.

•	The quantitative results for Preservation VLM, Edit Accuracy, CLIP, and LPIPS show that the performance of LoRA and LoRBA are quite close. Similarly, in the qualitative comparisons in Figure 4, LoRA appears to produce visually comparable results. Given that the observed performance gap between LoRA and LoRBA is relatively small, the reviewer is uncertain about why the community utilize this architecture instead of LoRA?  What is the main reason that makes LoRBA better than LoRA?

**Questions:**

•	In Section 2, what type of transformations are inferred?

•	Figure 2 is missing to show the edit prompt in the process.

•	How is LoRBA conceptually and architecturally different than inspired work of Dravid’s?

•	Regarding the test data created with 18 community LoRAs, were the outputs manually reviewed or verified by human evaluators? How are quality of the results ensured?

•	Table 1 requires a clearer explanation, as it is currently difficult to interpret without additional context or guidance.

•	The variations of capacity effect evaluation, such as {N = 32, r = 4}, are missing or not described well in Table 1.

•	Figure 7 is missing the results of LoRA.

•	What are the general tasks such as style transfer, background replacements, object insertion, object displacement etc. that LoRBA fails to generate accurate results and what might be the reason for that?

•	While the paper is motivated by the concept of visual analogy, the use of explicit edit prompts (e.g., “Turn this photo into an architectural rendering”) defines the transformation in advance and bypasses the reasoning process of visual analogy. It would be valuable to include an experiment without explicit transformation prompts to assess whether the architecture itself can work visual analogy task by inferring and applying the relation purely from the exemplar pair.

•	One of the related works using the same exemplary-based image editing method with LoRA is PairEdit. An additional experiment can be conducted to evaluate PairEdit on the same test set and compare its results with LoRBA?
PairEdit : https://arxiv.org/abs/2506.07992

---

### Official Review · Reviewer_DgSv · 2025-10-28

**Soundness:** 2
**Presentation:** 1
**Contribution:** 2
**Rating:** 2
**Confidence:** 3

**Summary:**

This paper addresses the task of visual analogy learning. The authors propose a two-stage framework: first, they train a set of LoRA modules alongside a corresponding set of learnable combination weights. In the second stage, these LoRA weights are combined to generate an image that fulfills the analogical relationship.

**Strengths:**

**High-Quality Visual Results:** The method produces results of high visual quality that demonstrate strong adherence to the analogical prompts when compared against baseline methods.

**Weaknesses:**

**Lack of Methodological Clarity:** The core weakness of this manuscript is the clarity of the methodology section. The description is ambiguous and lacks a clear, end-to-end overview of the training and inference processes. Furthermore, the mathematical notation is inconsistent and potentially confusing (e.g., the relationship between `a/b` as inputs and `A/B` as LoRA weights make reader confusing), which hinders a complete understanding of the proposed technique.

**Questions:**

A comprehensive evaluation of this work is contingent upon a clear understanding of the methodology. Could the authors please provide significant clarification on the following points?

1. **Elucidation of the Training and Inference Pipeline:** The current description of the pipeline is difficult to follow.

    - **Recommendation:** To resolve these ambiguities, I strongly recommend that the authors include a detailed diagram or, ideally, a **pseudo-code algorithm** that explicitly outlines both the complete training and inference procedures.

    - Given a single training instance with three inputs (`a`, `a'`, and `b`), could you please detail the process used to train the full set of N=32 LoRAs? The mapping from one training example to a large set of distinct LoRAs is not intuitive.

    - How are the learnable combination weights (`e_i` in Equation 4) incorporated into the training loss and updated during the optimization process?

    - Why use the 2x2 grid as the Flux’s input?

2. **Compare with LLM-based Methods:** Recent approaches, such as those leveraging Visual Language Models (e.g., "Nano-Banana"), have also been applied to analogy tasks. Could you please discuss the comparative advantages and potential limitations of your LoRA composition framework relative to these LLM-based methods? A discussion on aspects like inference speed, training cost, would be insightful.

---

### Official Review · Reviewer_BMgC · 2025-11-01

**Soundness:** 3
**Presentation:** 3
**Contribution:** 3
**Rating:** 4
**Confidence:** 4

**Summary:**

This paper proposes LoRBA, a visual-analogy editing method that replaces the single-LoRA adapter common in prior work with a learnable basis of LoRAs. A small encoder (frozen CLIP + projection) embeds the analogy triplet \{a,a’,b\} and routes softmax mixing coefficients over the basis to create a task-specific “mixed LoRA” at inference time; the diffusion backbone (FLUX.1-Kontext) receives the full triplet via extended attention while CLIP features are used only for LoRA selection.
Experiments on Relation252k (train) and a custom validation split show improved trade-offs between edit accuracy and preservation (VLM-based) and higher pairwise win rates vs. several strong baselines; ablations support design choices (basis size, softmax vs. tanh, CLIP routing input).

**Strengths:**

The paper motivates that a single LoRA under-represents the space of transformations, while a learned basis + router can specialize per analogy at inference. This is grounded in prior observations that LoRAs can span a semantic space.

LoRBA pushes the Pareto front on VLM edit-accuracy vs. preservation and wins user-study & VLM pairwise comparisons vs. most of baselines, indicating better edits without sacrificing identity.

**Weaknesses:**

Three prior-art baselines run on FLUX.1-Dev, whereas LoRBA (and a capacity-matched single-LoRA baseline) run on FLUX.1-Kontext. This makes it hard to attribute all gains solely to the LoRA-basis design rather than backbone differences. A fairness note is warranted or re-runs on the same backbone are needed.

Because Relation252k’s test set is unavailable, the authors build their own validation suite (Unsplash images, LLM-generated prompts, and community LoRAs). While thoughtfully constructed, this pipeline can encode distributional choices that favor the method; public release and stress tests would help.

Both scalar metrics (edit-accuracy & preservation) and pairwise selection use Gemma-3 prompts. Although there is a user study, the paper would benefit from reporting VLM–human correlation and sensitivity analyses (prompt variants).

Experiments cap the long edge at 512 and focus on FLUX.1-Kontext; it’s unclear how the approach scales to higher resolutions or transfers to other diffusion backbones.

Typo: “mosiac” → “mosaic” in Fig. 3 caption. Please standardize notation (e.g., keep e_i for coefficients consistently across text/equations) and explicitly reference Eq. (2–4) where the “Mixed LoRA” is injected.

**Questions:**

Clarify whether the custom validation set (images, prompts, LoRA list) and code will be released to enable reproduction

---

### Official Review · Reviewer_6GPV · 2025-11-01

**Soundness:** 3
**Presentation:** 4
**Contribution:** 3
**Rating:** 8
**Confidence:** 3

**Summary:**

This paper introduces method to jointly train 1) multiple LoRA modules and 2) a learnable encoder that assigns the coefficients for each LoRA module for image editing with visual analogy pairs.The paper compares against previous baselines and reports state-of-the-art on quantitative (CLIP score similarity + VLM-as-a-judge) and qualitative (user study) metrics.The paper also runs a suite of ablations to identify the core of the improvement.

**Strengths:**

1. The paper is well written. I’m not too familiar with the topic, but was able to understand the motivation and the setup clearly.
2. Strong performance gains against the baselines. The model is better at making accurate edits while preserving the original image. The results are validated with qualitative and quantitative metrics.
3. Test-time inference is efficient. There is no need to train a separate module / separate set of coefficients for a new task at test-time. The images only need to go through a CLIP model to retrieve the query vector.

**Weaknesses:**

1. It is unclear whether the out-of-domain tasks are truly distinct from the training analogy types. From how it is mentioned, it seems like the authors only sampled from LoRA modules for samples where the base model was unable to make edits for. Does this mean these analogy types are disjoint from the training analogy types? Were there manual checks?
2. Related to 1, does not test the limit of generalization (lines 482-483). Are there specific analogy types in the authors’ validation set that the model performs better/worse on?
3. Limited analysis of scalability. Any experiments on increasing N? Is it because of the limited data? Would the performance plateau?

**Questions:**

1. It seems like the validation data from Gong et al. (RelationAdapter) have now been released (https://huggingface.co/datasets/handsomeWilliam/Relation252K-unseen/tree/main). Could you also validate your method on the validation set just to remove any confounding factor from constructing your own validation set from a different pipeline?

2. Nit-picky but were the images presented to the survey participants randomized in order to remove any positional bias?

---

### Author Response · Authors · 2025-11-14

We thank all reviewers for their thorough review and constructive feedback. After careful consideration, we have decided to withdraw the paper at this stage to integrate the feedback for the next revision.

Below, we briefly address the main comments:

# Performance for different analogy types
Reviewers 6GPV and 9xKs asked if the methods perform better for some analogy types. Indeed, all methods (ours included) handle style-transfer tasks better than targeted edits (e.g., crown addition). The reason for this effect is as follows:
1. Text prompts provide more information for style-transfer relative to targeted edits.
2. Relation252K is style-transfer oriented, making methods trained on it struggle with significantly different tasks.
Tasks that the base model cannot generate are harder for all methods, but the relative success of LoRBA indicates that our LoRA basis can enable new editing capabilities for the base model.
# Experimental rigor and fairness
Reviewer 6GPV highlighted the recent partial release of Relation252K’s validation set and requested more scalability experiments. We will update our results accordingly.


# Uniqueness of LoRBA
Reviewer 9xKs asked how our approach differs from Dravid et al., and what improvements it offers over using a single LoRA. We emphasize that while our approach is inspired by Dravid et al, it differs from their work both conceptually and methodologically.
Dravid et al. train 65K different models, each on multiple face images of the same person, totalling in 65K different humans. They further curate this into a weight space using PCA, and then to use this space they require test-time optimization. This approach is resource demanding and data-hungry, both in training and in test time.
Our approach directly learns an efficient weight-basis with only 32 LoRAs and a single forward pass at inference. Additionally, while a single LoRA is a surprisingly strong baseline (which even outperforms some prior published work), LoRBA achieves consistently better results that preserve the fine details of the analogy.

# Prompts reliance
Reviewer 9xKs raised concerns about prompts bypassing visual reasoning.
The reviewer is correct that prompts help, Fig. 7 demonstrates how LoRBA does use the visual data in the analogy pairs. Unlike previous work which may  bypass content images and rely solely on the prompt.
Additionally, many prompts are simple and describe the general editing task type without specifying the exact details of the transformation. For example consider the bottom prompt in Figure 7  “Give this creature a crown of crystals”. This prompt  does not specify that the crown has a silver color and is  adorned with purple, diamond-shaped crystals at the top, separated by smaller gems, with circular gems along the base. Therefore, our method cannot bypass visual reasoning to successfully adhere to the analogy, and we find that it indeed uses the visual context.
We note that in LoRBA,  prompts are only used as an input to the base model, and LoRBA does not receive  textual input. As a result, one can train and apply LoRBA giving empty prompts to the base model, and even use a base model that does not accept prompts as input.
The prior methods use prompts as inputs. Prompts are an essential part of Relation252K on which LoRBA was trained, comparing prior art to a prompt-less version of LoRBA would not allow for a fair comparison.


# Distinctness of out-of-domain tasks
Reviewer 6GPV inquired regarding the community LoRAs, and asked whether the analogy types they represent are disjoint from the analogy types seen during training.
Indeed, the resulting edits from the community LoRAs differ from the types of edits seen during training. As mentioned in Line 311, we aim to test our model on analogies that are in/out-of-domain for the base model. For out-of-domain samples, we cannot use the base-model to generate context pairs, because, by definition, the model cannot perform these edits.
Since the training data of Flux is unknown, it is not clear how to construct an out-of-domain analogy test set. For this reason we opt for community LoRAs, which are generally trained to enable the base model to perform new types of edits. Here, we assume that these LoRAs were indeed trained as a result of the base model not producing the wanted transformation, since otherwise there is no point in fine-tuning these LoRAs. We manually verified the outputs differ from standard FLUX generation.


# Reproducibility
We commit to releasing the evaluation set and codebase upon acceptance.



We again thank the reviewers for their expertise and time given to our paper.

---

### Note · Authors · 2025-11-14

I have read and agree with the venue's withdrawal policy on behalf of myself and my co-authors.